# A microbial clock provides an accurate estimate of the postmortem interval in a mouse model system

Jessica L Metcalf[1]*, Laura Wegener Parfrey[1]†, Antonio Gonzalez[1], Christian L Lauber[2], Dan Knights[3,4], Gail Ackermann[1], Gregory C Humphrey[1], Matthew J Gebert[1], Will Van Treuren[1], Donna Berg-Lyons[1], Kyle Keepers[1], Yan Guo[5], James Bullard[6], Noah Fierer[2,7], David O Carter[8], Rob Knight[1,9,10,11]*

[1]Biofrontiers Institute, University of Colorado at Boulder, Boulder, United States; [2]Cooperative Institute for Research in Environmental Sciences, University of Colorado at Boulder, Boulder, United States; [3]Department of Computer Science and Engineering, University of Minnesota, Minneapolis, United States; [4]BioTechnology Institute, University of Minnesota, Saint Paul, United States; [5]Field Application Support, Pacific Biosciences, Menlo Park, United States; [6]Biostatistics, Pacific Biosciences, Menlo Park, United States; [7]Department of Ecology and Evolutionary Biology, University of Colorado at Boulder, Boulder, United States; [8]Laboratory of Forensic Taphonomy, Division of Natural Sciences and Mathematics, Chaminade University of Honolulu, Honolulu, United States; [9]Howard Hughes Medical Institute, University of Colorado at Boulder, Boulder, United States; [10]Department of Computer Science, University of Colorado at Boulder, Boulder, United States; [11]Department of Chemistry and Biochemistry, University of Colorado at Boulder, Boulder, United States

*For correspondence: rob. knight@colorado.edu (RK); jessicalmetcalf@gmail.com (JLM)

†Present address: Departments of Botany and Zoology, University of British Columbia, Vancouver, Canada

**Abstract** Establishing the time since death is critical in every death investigation, yet existing techniques are susceptible to a range of errors and biases. For example, forensic entomology is widely used to assess the postmortem interval (PMI), but errors can range from days to months. Microbes may provide a novel method for estimating PMI that avoids many of these limitations. Here we show that postmortem microbial community changes are dramatic, measurable, and repeatable in a mouse model system, allowing PMI to be estimated within approximately 3 days over 48 days. Our results provide a detailed understanding of bacterial and microbial eukaryotic ecology within a decomposing corpse system and suggest that microbial community data can be developed into a forensic tool for estimating PMI.

## Introduction

Forensic science is concerned with identifying and interpreting physical evidence. Long-standing forms of physical evidence include fingerprints, bloodstains, hairs, fibers, soils, and DNA. However, any object can serve as physical evidence if it provides reliable insight into the activities associated with a death scene or crime. Physical evidence is often of critical importance to a criminal investigation because testimonial evidence, the statements provided by victims, suspects, and witnesses, is frequently incomplete and inaccurate (*Saks and Koehler, 2005*). To address these limitations of testimony, an investigator will compare the statements to the interpretation of physical evidence. For example, an investigator will require a suspect to provide an alibi to describe his or her whereabouts during the time period of a crime and compare their statement to mobile phone activity.

**eLife digest** Our bodies—especially our skin, our saliva, the lining of our mouth and our gastrointestinal tract—are home to a diverse collection of bacteria and other microorganisms called the microbiome. While the roles played by many of these microorganisms have yet to be identified, it is known that they contribute to the health and wellbeing of their host by metabolizing indigestible compounds, producing essential vitamins, and preventing the growth of harmful bacteria. They are important for nutrient and carbon cycling in the environment.

The advent of advanced sequencing techniques has made it feasible to study the composition of this microbial community, and to monitor how it changes over time or how it responds to events such as antibiotic treatment. Sequencing studies have been used to highlight the significant differences between microbial communities found in different parts of the body, and to follow the evolution of the gut microbiome from birth. Most of these studies have focused on live animals, so little is known about what happens to the microbiome after its host dies. In particular, it is not known if the changes that occur after death are similar for all individuals. Moreover, the decomposing animal supplies nutrients and carbon to the surrounding ecosystem, but its influence on the microbial community of its immediate environment is not well understood.

Now Metcalf et al. have used high-throughput sequencing to study the bacteria and other microorganisms (such as nematodes and fungi) in dead and decomposing mice, and also in the soil beneath them, over the course of 48 days. The changes were significant and also consistent across the corpses, with the microbial communities in the corpses influencing those in the soil, and vice versa. Metcalf et al. also showed that these measurements could be used to estimate the postmortem interval (the time since death) to within approximately 3 days, which suggests that the work could have applications in forensic science.

One of the most difficult forms of physical evidence to establish is the amount of time that has lapsed since death, the postmortem interval (PMI). Establishing the PMI is critical to every death investigation because it facilitates the identification of victims and suspects, the acceptance or rejection of suspect alibis, the distribution of death certificates, and the allocation of assets outlined in wills. However, PMI is difficult to establish because we have a relatively poor understanding of corpse decomposition. To improve the ability to estimate PMI, forensic science has incorporated an ecological perspective. When an animal dies it becomes a large nutrient resource that can support a complex and phylogenetically diverse community of organisms (*Mondor et al., 2012*). As this decomposer community recycles nutrients, the corpse progresses through several forensically recognized stages of decomposition, including Fresh (before decomposition begins), Active Decay, which includes Bloating and Rupture, and Advanced Decay (*Carter et al., 2007*; *Parkinson et al., 2009*). Biotic signatures associated with these stages of decomposition, such as the development rate of blow fly larvae (*Amendt et al., 2007*), succession of insects (*Horenstein et al., 2010*), and changes in the biochemistry of corpse-associated 'gravesoil' (*Tullis and Goff, 1987*; *Vass et al., 1992*; *Benninger et al., 2008*; *Carter et al., 2008*; *Horenstein et al., 2010*), can be used to estimate PMI. However, no method is successful in every scenario (*Tibbett and Carter, 2008*). For example, limitations of forensic entomology include uncertainty in the interval between death and egg deposition (*Tomberlin et al., 2011*), lack of insects during particular weather events or seasons (*Archer and Elgar, 2003*), and region-specific blowfly larval growth curves and insect communities (*Gallagher et al., 2010*). Using microbial community change to track the progression of decomposition may circumvent many of these limitations because microbes are ubiquitous in the environment, located on humans before death, and can be reliably quantified using high-throughput DNA sequencing.

Microbes play an important role in decomposition (*Vass, 2001*; *Hopkins, 2008*; *Mondor et al., 2012*). For example, from the Fresh stage to the Bloat stage, enteric microbes likely contribute to putrefaction by digesting the corpse macromolecules, which in turn generates metabolic byproducts that cause the corpse to bloat (*Mondor et al., 2012*). Evans proposed that a major shift in microbial communities occurs at the end of bloat when the body cavity ruptures (*Evans, 1963*), as this key event likely shifts the abdominal cavity from anaerobic to aerobic. Additionally at the Rupture stage, nutrient rich body fluids are released into the environment often increasing pH (*Carter et al., 2010*) likely

altering endogenous microbial communities. The microbiology of corpse decomposition can now be investigated in detail by utilizing sequencing advances that enable entire communities to be characterized across the timeline of decomposition. These data will not only allow us to understand the underlying microbial ecology of corpse decomposition, but also the feasibility of using microbes as evidence. We must rigorously test whether microbial community change is sufficiently measurable and directional during decomposition to allow accurate estimates of past events such as the PMI. Recently, *Pechal et al. (2013)* demonstrated that skin and mouth bacterial communities followed a consistent trajectory of change during decomposition of three swine corpses over 5 days. Here, we expand upon these initial promising results by characterizing microbial community change in three external sites and one invasively sampled internal site in 40 destructively sampled mouse corpses over 48 days. In addition to characterizing bacterial community change of skin samples, we provide the first high-throughput sequencing characterization of microbial communities in the abdominal cavity and corpse-associated soils. Furthermore, we present the first high-throughput sequencing characterization of microbial eukaryotic community change (e.g., fungi, nematodes, and amoeba) associated with skin sites, the abdominal cavity and corpse-associated soils.

We conducted a laboratory experiment to characterize temporal changes in microbial communities associated with mouse corpses as they decomposed on soil under controlled conditions for 48 days. We characterized bacteria, archaea, and microbial eukaryotes using a combination of culture-independent, high-throughput DNA sequencing approaches for each sample. To characterize the composition and diversity of bacterial and archaeal communities, we extracted DNA and sequenced partial 16S ribosomal RNA (rRNA) genes for each sample. To characterize the microbial eukaryotic community of each sample, we sequenced partial 18S rRNA genes. To accurately characterize these highly diverse microbial communities, we sequenced ~100 base pairs of both 16S and 18S amplicons at a depth of millions of sequences using the Illumina HiSeq platform. In order to gain species and genus level taxonomic resolution of key taxa, a much longer fragment of the rRNA gene (roughly 800 base pairs for 16S amplicons and 1200 base pairs for 18S amplicons) was also sequenced at a level of thousands of sequences for a subset of early and late stage decomposition samples on the Pacific Biosciences RS platform. Together, these data allowed us to assess hypotheses long-held by the forensic community about the role of microbes in corpse decomposition, and rigorously test whether changes in microbial communities are predictable over the timeline of decomposition, which is crucial for assessing whether microbes can be used as a 'clock' to estimate PMI.

## Results and discussion

### Study design and data generation

We sampled the microbial communities from the mouse corpse abdominal cavity and skin (head and torso), as well as from the associated gravesoil from five replicate corpses over eight time points that spanned 48 days (*Table 1*, *Supplementary file 1A*). By using a mouse model, we were able to perform a highly replicated experiment with destructive sampling that enabled us to sample both the surface and interior of each corpse at each sampling time point. Importantly, this approach facilitated access to the abdominal cavity prior to natural corpse rupture, and thus our data addresses forensic hypotheses that the abdominal microbes play a key role in corpse decomposition (*Evans, 1963*). This approach contrasts with typical forensic studies of decomposition where one to three donated corpses of human or swine (used as a human model) are used for experiments where only externally accessible body sites are sampled (e.g., *Pechal et al., 2013*). A further advantage of using a mouse model system is that the large number of samples allowed us to assess to what extent the intra-individual variation of microbiota that we know is present in living humans and other mammals (*Costello et al., 2009*) extends to microbial community change during decomposition.

Over the duration of our 48-day experiment, mice progressed through all major stages of decomposition (*Figure 1*). We assessed the progression of decomposition in two ways. First, we recorded a visual body score estimate for the head and torso according to *Megyesi et al. (2005)* at the time samples were collected. Visual body score estimates suggested that day 0 and 3 primarily included mice considered in the Fresh stage. Days 6, 9, and 13 included mouse corpses that were primarily scored as Active Decay, including the Bloat stage on ~days 6–9. Finally, mice on days 20, 34, and 48 were primarily categorized as Advanced Decay (*Figure 1A*). Second, after the conclusion of the experiment, we measured the pH of soil samples to estimate the timeframe during which the purging of

**Table 1.** Total number of samples collected for each site and abdominal, skin (head and body), soil (with and without corpses)

| Sample type | Samples collected | Samples sequenced (16S HiSeq) | Samples sequenced (18S Hiseq) | Samples sequenced (PacBio) |
|---|---|---|---|---|
| Abdominal | 65 | 43 | 23 | 12 |
| Skin of body | 53 | 33 | 31 | 0 |
| Skin of head | 40 | 36 | 29 | 6 |
| Soil with corpse | 53 | 46 | 60 | 8 |
| Soil no corpse | 12 | 9 | 8 | 0 |
| Sum | 223 | 167 | 152 | 26 |

We show the number of successfully sequenced samples for each data type, including Illumina Hiseq and Pacific Biosciences. For Illumina data, we only included samples used in statistical analyses, which required>2500 sequences/sample. Details about each of the individual samples can be found in **Supplementary file 1A**.

fluids or Rupture had likely occurred. It is well established that decomposition fluids are released from a body during decomposition (**Carter et al., 2007**). These fluids are a pulse of nutrients that increase the pH of the surrounding soil. Measurements of soil pH indicated that Rupture (Active Decay) occurred between days 6 and 9 of the experiment (**Figure 1B**). Overall, we concluded that the mice were in the Fresh stage on approximately days 0–3, the Active Decay stage on days 6–20 (with Bloat primarily occurring after day 3 until day 9 and Rupture occurring after day 6 and before day 9) and the Advanced Decay on days 20–48. These stage classifications were used in subsequent analyses.

In total, we collected 223 abdominal cavity, skin, corpse-associated soils samples (gravesoil), and no-corpse soil controls. After sequence quality filtering and removal of failed samples and samples with low numbers of sequences, the HiSeq Illumina sequence dataset included 167 samples and 2,931,901 16S rRNA sequences that represented 4505 OTUs (**Table 1**, **Supplementary file 1A**). After sequence and sample filtering, the 18S dataset included 142 samples and 21,254,848 18S rRNA sequences that represented 421 OTUs (**Table 1**, **Supplementary file 1A**). Most samples with too few sequences to be included in the final datasets (i.e., failures) were collected at early time points (e.g., days 0 and 3) when microbial biomass was likely low. This was especially true for eukaryotic skin samples and abdominal cavity swab or liquid samples (although fecal and cecum day 0 samples worked well as expected for these rich microbial habitats). The 16S and 18S Illumina HiSeq datasets were used for downstream statistical analyses (e.g., Mantel tests, PERMANOVA tests, and PMI estimates). Finally, for better taxonomic resolution via longer sequence reads, we also sequenced 26 samples using Pacific Biosciences sequencing platform resulting in a total of 16,250 sequences for 18S and 16S. These data were used separately from the Illumina data to identify highly abundant taxa present during early and late stage decomposition at the genus and species levels (e.g., the nematode species *Oscheius tipulae*).

## The ecology of bacterial community change during decomposition

It has long been assumed that endogenous gut-associated bacteria dominate cadaver decomposition prior to rupture, following which non-enteric and aerobic (soil-borne, dermal) microbes bloom and dominate the community (**Evans, 1963**). Rupture is a crucial stage during decomposition, in which bloating due to putrefaction breaks open the abdominal cavity, and is expected to result in shifts of the microbial community because the cavity becomes aerobic. Early culture-based investigations conducted without soil lent support to this assumption by showing that many bacteria exploiting a carcass are members of the gut microbiota (**Ingram and Dainty, 1971**; **Corry, 1978**). Our results also lend support to these long-held hypotheses. During the Bloating Stage (approximately days 6–9), endogenous anaerobes and facultative anaerobes that are known to be common members of the gut community such as Firmicutes in the families Lactobacillaceae (e.g., *Lactobacillus*) and Bacteroidetes in the family Bacteroidaceae (e.g., *Bacteroides*) (**Supplementary file 1B**) increase in the abdominal cavity. However, after rupture occurs (~9 days after the start of the experiment), these taxa decrease dramatically, and exposure of the abdominal cavity to oxygen allows aerobes such as members of the

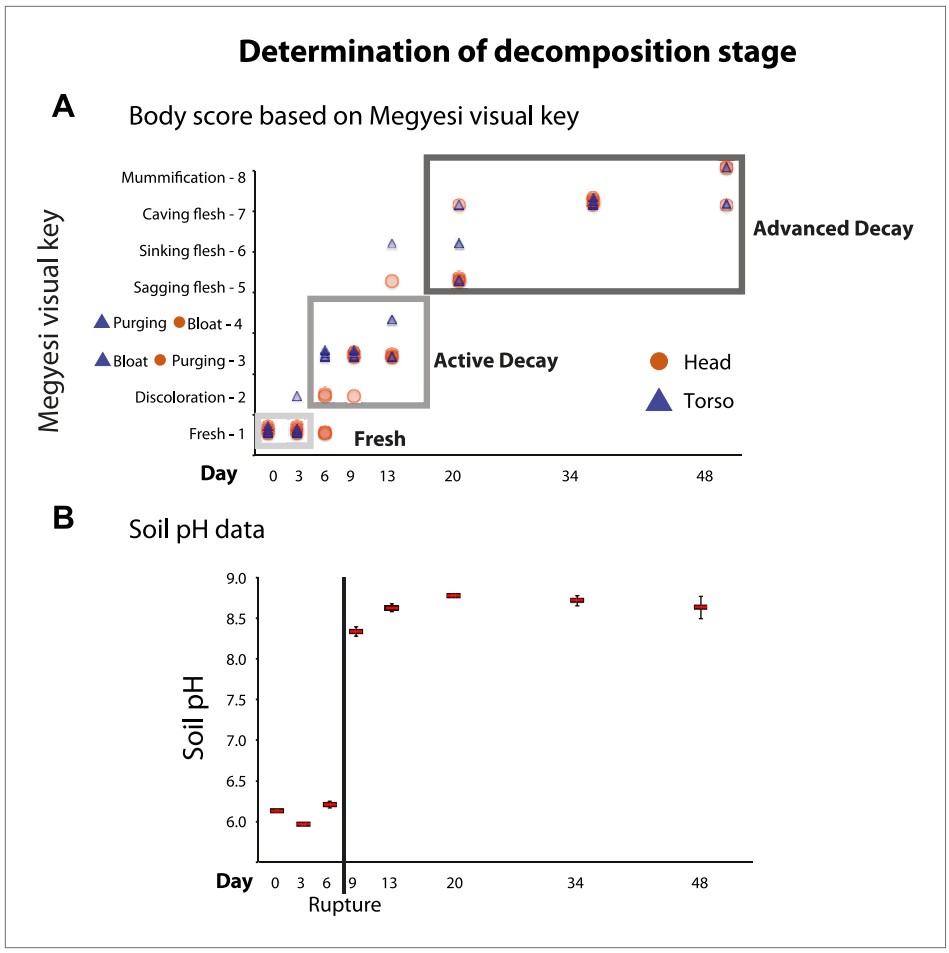

**Figure 1**. We used two independent methods to assess stages of decomposition: (A) a visual body score estimate of decomposition following the Megyesi Key (***Megyesi et al., 2005***) and (B) the pH of soil to determine when rupture had occurred. (**A**) visual key estimates for the head (orange circle): **Fresh**–no discoloration (1 point); **Active Decay**–Discoloration (2 points), Purging of decomposition fluids out of eyes, nose, or mouth (3 points), Bloating of neck and/or face (4 points); **Advanced Decay**–Sagging of flesh (5 points), Sinking of flesh (6 points), Caving in of flesh (7 points), Mummification (8 points). The key for the torso (blue triangle) is the same as above except that Bloating of abdominal cavity (3 points) precedes Rupture and/or purging of fluids (4 points). Gray boxes around points indicate generally with which stage of decay each time point is associated–Fresh (~days 0, 3), Active Decay (~days 6, 9, 13), and Advanced Decay (~days 20, 34, 48). (**B**) Average pH of soil over time with standard error. A dramatic increase in pH occurred between day 6 and day 9, which is when rupture of body fluids and subsequent leakage into the soil likely occurred.

Rhizobiales (Alphaproteobacteria) in the families Phyllobacteriaceae, Hyphomicrobiaceae, and Brucellaceae (e.g. *Pseudochrobactrum* and *Ochrobactrum*) to dominate (***Figure 2A,B***, ***Supplementary file 1B***). Additionally, facultative anaerobes in the Gammaproteobacteria family Enterobacteriaceae such as *Serratia*, *Escherichia*, *Klebsiella*, and *Proteus* (***Supplementary file 1B***), which are widely recognized as opportunistic pathogens and are associated with sewage and animal matter (***Leclerc et al., 2001***), become abundant after rupture.

When corpses rupture they release an ammonia-rich, high nutrient fluid that alters both the pH and nutrient content of the soil (***Meyer et al., 2013***). Accordingly we saw a predictable spike in gravesoil pH from ~6.0 to ~8.5 (***Figure 1B***) and declines in Acidobacteria (***Figure 2A***), the abundance of which is known to be inversely related to soil pH (***Lauber et al., 2009***). Acidobacteria prefer oligotrophic conditions (***Fierer et al., 2007***) and grow much more slowly than most other taxa (***Ward et al., 2009***; ***Castro et al., 2010***), thus the decline of Acidobacteria may be related to the huge pulse of nutrients into the soil rather than shifts in soil pH. In this study, Alphaproteobacteria abundance (mainly the

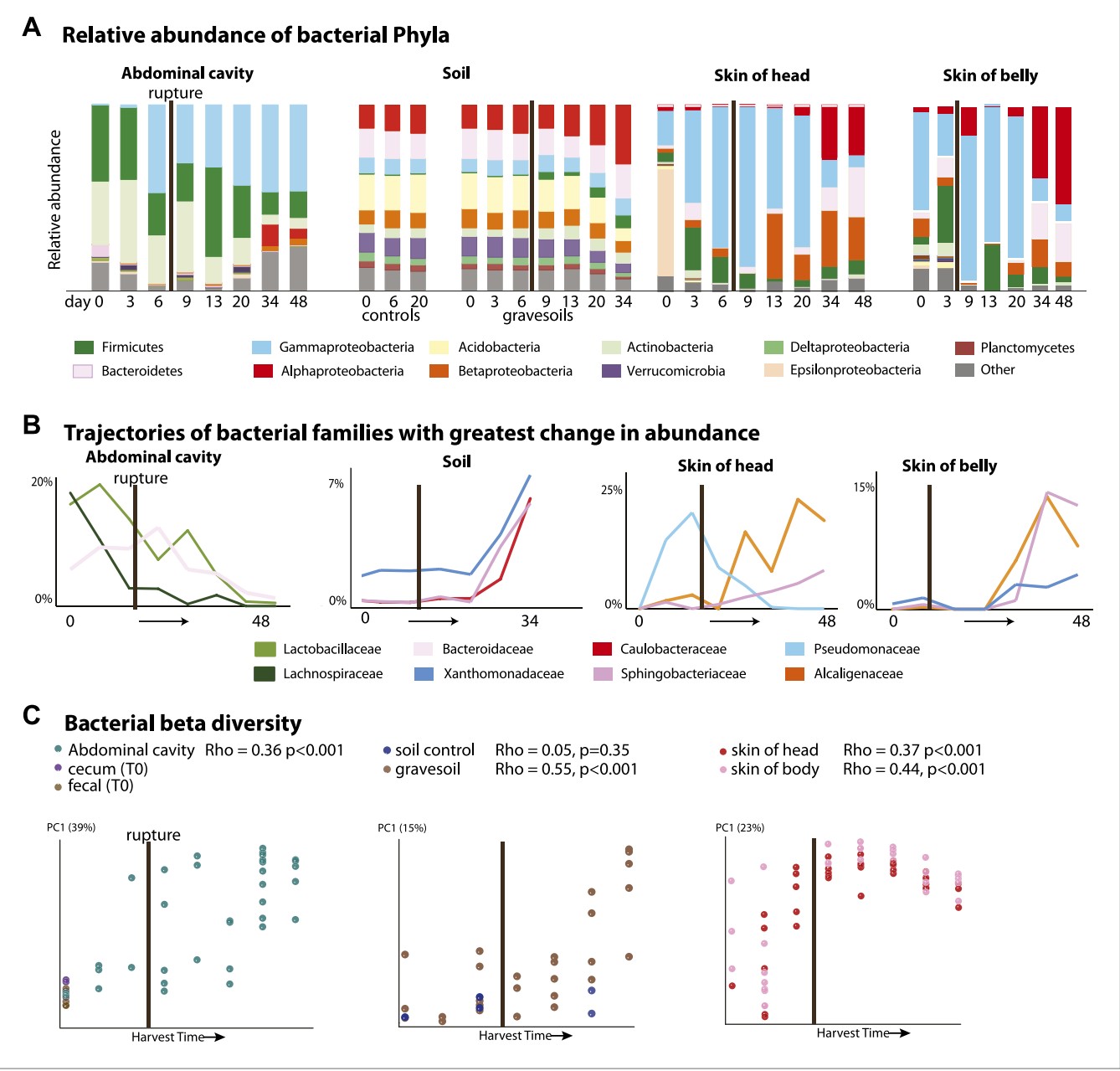

**Figure 2**. Bacterial community composition changes significantly and consistently over the course of decomposition. (**A**) Relative abundance of phyla of bacteria over time for all body sites. For the abdominal site, Day 0 includes cecum, fecal, and abdominal swab and liquid samples. For the soil site, control soils collected on days 0, 6, and 20 are shown on the left of the plot. (**B**) The three bacterial families that show the greatest change in abundance over time are plotted for each site. (**C**) PCoA plot based on unweighted UniFrac distances displaying bacterial community change at all sites during decomposition. Results from Mantel tests (Rho and p values) show that bacterial community change correlated significantly with time. The point of rupture is marked with a thick vertical black line on each plot.

Rhizobiales) increased across the sampling period and became most abundant post-Rupture stage soil samples (**Table 2**), suggesting this group prefers a relatively nutrient rich environment and is able to outcompete the Acidobacteria (**Marilley and Aragno, 1999**; **Klappenbach et al., 2000**; **Smit et al., 2001**) for the corpse derived resources. Such shifts in taxa composition were typical in these Advanced Decay associated soil communities (after ~20 days), which were significantly different from the time-zero soil communities as well as the no-corpse soil controls (**Figure 2A**, **Table 3**, PERMANOVA p value 0.007). No-corpse control soils, which were collected throughout the experiment, did not change significantly

**Table 2.** The five most changing bacterial taxa groups resolved to the family level for each site over the timeline of the experiment based on HiSeq Illumina results

| Rank change | Soil | Abdominal | Skin |
|---|---|---|---|
| 1 | **Gammaproteobacteria Xanthomonadales Xanthomonadaceae** | **Clostridia Clostridiales Clostridiaceae** | Epsilonproteobacteria Campylobacterales Campylobacteraceae |
| 2 | **Alphaproteobacteria Caulobacterales Caulobacteraceae** | Bacilli Lactobacillales Lactobacillaceae | Gammaproteobacteria Pseudomonadales Pseudomonadaceae |
| 3 | **Sphingobacteria Sphingobacteriales Sphingobacteriaceae** | Bacteroidia Bacteroidales Bacteroidaceae | **Betaproteobacteria Burkholderiales Alcaligenaceae** |
| 4 | **Alphaproteobacteria Rhizobiales Aurantimonadaceae** | Clostridia Clostridiales Lachnospiraceae | **Sphingobacteria Sphingobacteriales Sphingobacteriaceae** |
| 5 | **Alphaproteobacteria Sphingomonadales Sphingomonadaceae** | **Bacilli Lactobacillales Enterococcaceae** | **Gammaproteobacteria Xanthomonadales Xanthomonadaceae** |

Groups that increase in abundance are listed in bold text and groups that decreased in abundance are shown in normal text.

over time (Mantel Rho 0.05, p value 0.35), linking soil community change to the presence of a corpse. Though preliminary, our work suggests that it may be possible to identify gravesoil by an increase in the abundance of copiotrophic taxa relative to oligotrophic taxa.

Bacterial communities associated with the decomposing corpses became increasingly differentiated from starting communities over time in the abdominal cavity, gravesoil, and skin sites (*Figure 2*, *Table 4*, Mantel test Rho values 0.36–0.55, p<0.001 for all sites). Although they did not converge completely, similar taxa became abundant at each sample site in the later stages of decomposition (*Supplementary file 1C*—weighted UniFrac distances between sample sites during Advanced Decay were significantly lower than between sites during the Fresh stage, two sample *t* test <0.000001 for each of five comparisons). For example, in both soil and skin, several bacterial families within the Bacteroidetes (Sphingobacteriaceae), Alphaproteobacteria (Brucellaceae, Phyllobacteriaceae, and Hyphomicrobiaceae), and Betaproteobacteria (Alcaligenaceae) increase in abundance during the Advanced Decay stage of decomposition (*Figure 2A,B*, *Table 2*). This trend is consistent with previous findings that bacterial skin communities are often a reflection of the surrounding environment with

**Table 3.** For each sample site and each marker type, PERMANOVA results of UniFrac distance (unweighted) for Fresh (day 0–3 days) vs Advanced Decay (days 20–48) decomposition microbial communities

| | PERMANOVA pseudo F | p value (999 permutations) |
|---|---|---|
| 16S soil with corpse | 2.38 | 0.007 |
| 16S ctrl soil vs Advanced Decay soil | 2.54 | 0.001 |
| 16S abdominal | 6.31 | 0.001 |
| 16S skin on head | 8.19 | 0.001 |
| 16S skin on body | 5.81 | 0.001 |
| 18S soil with corpse | 10.17 | 0.001 |
| 18S ctrl soil vs Advanced Decay soil | 5.23 | 0.001 |
| 18S abdominal | 5.34 | 0.001 |
| 18S skin on head | – | – |
| 18S skin on body | 5.96 | 0.001 |

For soil sites, we also include comparisons of control soils vs Advanced Decay gravesoils. For the 18S skin of head, there were not sufficient samples for statistical analysis.

**Table 4.** For each sample site and each marker type, Mantel test results using Spearman's rank correlation coefficient to assess the correlation between microbial community UniFrac distance (unweighted) and time

|  | Spearman Rho | Spearman p value |
|---|---|---|
| 16S soil | 0.548 | 0.001 |
| 16S ctrl soil | 0.051 | 0.352 |
| 16S abdominal | 0.364 | 0.001 |
| 16S skin of head | 0.368 | 0.001 |
| 16S skin of body | 0.437 | 0.001 |
| 18S soil | 0.772 | 0.001 |
| 18S ctrl soil | 0.127 | 0.154 |
| 18S abdominal | 0.209 | 0.029 |
| 18S skin of head | 0.279 | 0.004 |
| 18S skin of body | 0.079 | 0.143 |

Importantly, control soil microbial communities did not change significantly over time.

which they are in contact (*Costello et al., 2009*; *Song et al., 2013*), and this convergence may also arise because the low biomass initially found on the skin is easily overwhelmed by soil taxa.

## The ecology of microbial eukaryotic community change during decomposition

The community of microbial eukaryotes also changed significantly and consistently over the time course of decomposition at all sampled sites except the skin of the torso (*Figure 3A*, *Table 4*). Beginning at approximately 20 days, the microbial eukaryotic community at all sites became dominated by a nematode, *O. tipulae*, in the family Rhabditidae (*Figure 3B*, *Tables 2 and 5*). Microbial eukaryotic community composition in the no-corpse control soils did not change significantly over time (correlation with time: Mantel Rho 0.15, p=0.13), though diversity did decline in the controls (*Figure 4*). Control soils were significantly different from soils associated with corpses in Advanced Decay (PERMANOVA pseudo-F 5.23, p value 0.001). Furthermore, the nematode *O. tipulae* was not detected at a level of >1% in any control soil sample.

*O. tipulae*, a bacterivorous representative of the family Rhabditidae, is considered a common nematode species of terrestrial habitats such as soil, leaf litter, and compost all over the world (*Baille et al., 2008*). As a consequence of the nematode bloom, Shannon diversity (community evenness) declines at all sample sites for eukaryotic communities (*t* test Fresh Stage vs Advanced Decay stage decomposition: soil p<0.001, abdominal p=0.002, skin p=0.03). Phylogenetic distance diversity estimates were variable across sample sites with a significant decrease only detected in soil (*Figure 4*). The nematode bloom is decoupled from rupture, and it appears that this generalist consumer of bacteria responds to the increase in bacterial biomass that is associated with decomposition (*Benninger et al., 2008*; *Carter et al., 2008*, *2010*; *Parkinson et al., 2009*; *Damann et al., 2012*) and outcompetes other community members. One potential contributing factor to the nematode bloom may have been the fact that the mouse graves were relatively closed systems that would have prevented entry of organisms preying on nematodes. Future experiments with open systems will be necessary to determine the impact of higher trophic levels on community dynamics.

## A microbial clock provides an accurate estimate of PMI

Because consistent shifts in the presence and abundance of specific bacterial and eukaryotic taxa occurred during known stages of decomposition, these data suggested that succession of bacterial and microbial eukaryotic communities may be used to estimate PMI. By regressing known postmortem interval directly on the taxon relative abundances using a Random Forests model. Random Forests is a machine learning technique that creates random decision trees based on subsets of the features (e.g., taxa) in the data, then chooses the subsets of features that are best able to classify samples into predefined groups using only part of the data. This technique has been useful in other microbial ecology studies, providing high classifier accuracy (*Knights et al., 2011*). We discovered that the temporal change in microbial communities of the skin of the head allowed us to estimate PMI within as little as 3.30 +/− 2.52 days (mean absolute error +/− standard deviation) (*Figure 5*). Regressions performed for the timeframe of 0–34 days resulted in the smallest mean absolute error (*Supplementary file 1D*), which suggests that microbes may be more highly informative for estimating PMI during the earlier stages of decomposition, at least for the sites we sampled. However, this trend may also be due to less frequent sampling events between days 13–48. It may be possible to further improve error estimates with more frequent sampling.

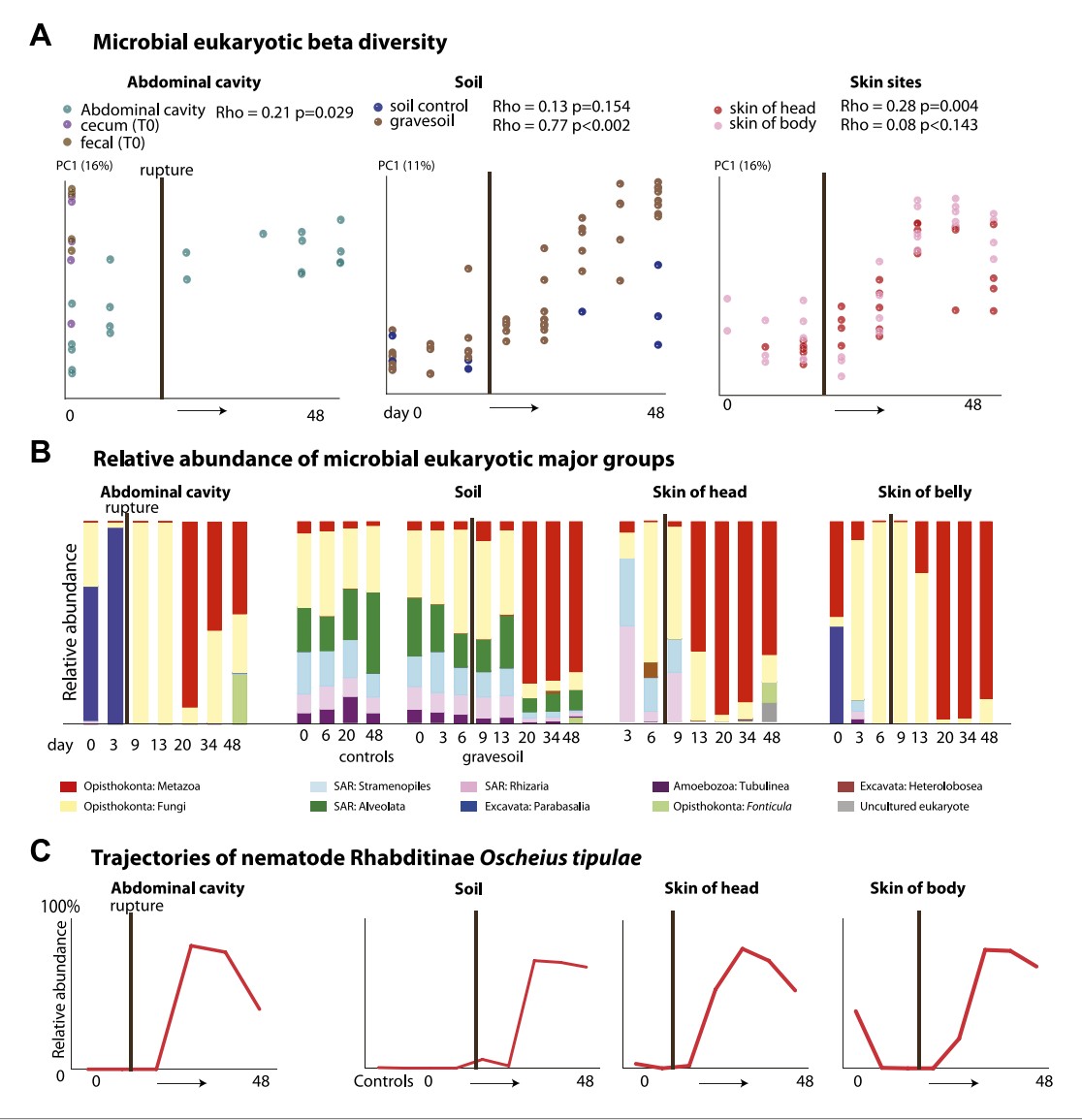

**Figure 3**. Eukaryotic community composition changes directionally and becomes dominated by the nematode *Oscheius tipulae*. (**A**) PCoA plot based on unweighted UniFrac distances displaying microbial eukaryotic community change during decomposition. Results from Mantel tests (Rho and p values) show that microbial eukaryotic community change correlated significantly with time except for the skin of the belly. (**B**) Relative abundance of microbial eukaryote taxa at the class level over time. Microbial eukaryotic community composition changes significantly and predictably over the course of decomposition. (**C**) The eukaryotic nematode *O. tipulae* became highly abundant at each sample site at late stages of decomposition. The point of rupture is marked with a thick vertical black line on each plot.

For both 16S and 18S datasets, skin and soil sites were more informative for estimating PMI than data from the abdominal cavity (*Figure 5*, *Supplementary file 1D*). This observation was initially surprising because the marked shifts from anaerobic to aerobic communities that occurred in the abdomen would appear to be a good marker of timing. However, these major microbial changes, which correspond to the time of Rupture, occurred at different times for the replicate corpses in the experiment (*Figures 2, 3 and 6A*), and this variability likely reduced the usefulness of the abdominal data for estimating PMI.

Overall, bacteria did not perform significantly better in our PMI estimates than microbial eukaryotes (*Supplementary file 1D*; two tailed, Wilcoxon signed rank test) and combining both 16S and 18S datasets resulted in the best estimates of PMI (*Figure 5*, *Supplementary file 1D*), although the improvement was not significant (*Supplementary file 1D*). For each sample site, we estimated the

**Table 5.** The five most changing microbial eukaryotes for each site over the timeline of the experiment based on HiSeq Illumina results

| Rank change | Soil | Abdominal | Skin |
|---|---|---|---|
| 1 | Rhabditinae *Oscheius* | Rhabditinae *Oscheius* | Rhabditinae *Oscheius* |
| 2 | Pythiaceae *Pythium* | Tritrichomonas | Rhizaria |
| 3 | Rhizaria | **Zygosaccharomyces** | Mucoraceae *Rhizomucor* |
| 4 | **Alveolata** | Ascomycota **Graphium** | Chromulinaceae *Uroglena* |
| 5 | **Zygomycetes** | **Nucleariidae Fonticula** | Pythiaceae Pythium |

Groups that increase in abundance are listed in bold text and groups that decreased in abundance are shown in normal text.

'importance' of each taxon, a measure of its contribution to the PMI estimate, by removing each taxon from the predictive model and calculating the mean percent increase in mean squared error (*Supplementary file 1E*). For bacteria, taxa in the Order Rhizobiales were among the most important predictive taxa at each sample site. For microbial eukaryotes, *Oscheius* was the most important taxon for skin and abdominal sites and the second most important for gravesoil. Fungal taxa of the family Boletales were the most important contributors to PMI regressions for gravesoil. An analysis of models built using small numbers of highly predictive taxa indicates that 5–10 taxa provide approximately the same prediction accuracy as models built with all taxa, as assessed by leave-one-out analyses (*Figure 6B*). This holds across domains and sample sites, and identifying diagnostic taxa may be a fruitful avenue for forensic investigation. Future studies involving varied habitat conditions, soil types, etc are needed to identify the most general subset of predictive taxa.

In all regression analyses, we directly evaluated the predictive strength of the fitted models by holding out one sample, fitting a regression using the remaining $n–1$ samples, and then using the fitted model to predict PMI for the held out sample. This is a more rigorous inference task than simply fitting a regression to all points jointly; it directly assesses the ability of the model to generalize to data it has not seen before. This distinguishes our approach from prior work in which regression models were evaluated by their ability to explain variation in all data points, after being fit to all data points (*Pechal et al., 2013*).

We have provided a proof-of-principle demonstration that microbial community ecology of decomposing corpses has potential to be developed into a complementary forensics tool for estimation of PMI. Additionally, we have provided the first bacterial high-throughput sequence time-series dataset for the abdominal cavity and gravesoil as well as the first microbial eukaryotic high-throughput sequence time-series dataset for the abdominal cavity, gravesoil, and skin sites of a decomposing animal. We also demonstrated that combining complementary sequencing technologies yields high-resolution taxonomic data. Furthermore, the approach described here has several advantages that make widespread adoption of these techniques likely in the future, including continued rapid decline in the cost of DNA sequencing and the ease and familiarity of swab-based sampling at potential crime scenes to law enforcement officers. Given the accuracy of our estimates of PMI, our work suggests a potentially expanded role for microbiological forensic indicators within the criminal justice system. However, the degree to which microbial community changes are sensitive to factors including environmental and edaphic conditions, variation in the human microbiome, and differences among host species, remain to be established in future work. It is promising that *Pechal et al. (2013)* recently discovered a similar consistent change of skin bacterial communities on three decomposing swine in an outdoor setting. Future studies are crucial because previous forensic investigations using traditional indicators has shown that environmental parameters such as temperature, moisture, soil type, and soil texture can affect the rate at which decomposition occurs and the ability of decomposer organisms to function (*Carter et al., 2008*, *2010*). It is likely that these parameters will affect postmortem microbial communities. Consequently, the relationship between these parameters, decomposition, and postmortem microbial communities must be understood to develop a method to estimate PMI that is admissible as physical evidence within the criminal justice system. However, the present work strongly indicates that decomposition is sufficiently reproducible as an indicator of PMI to motivate these additional studies.

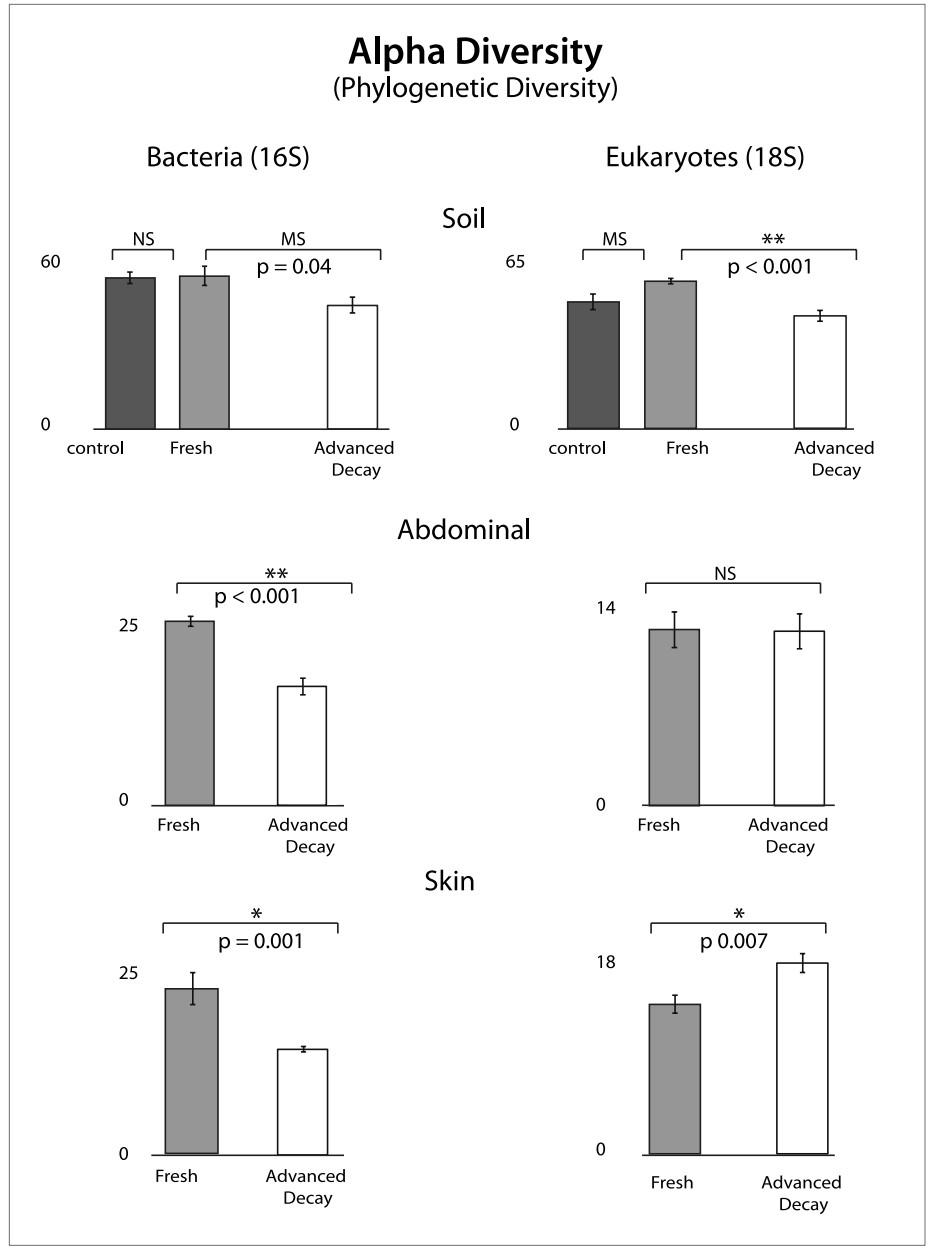

**Figure 4**. Phylogenetic distance (PD) alpha diversity and standard error for both bacterial (16S) and eukaryotic (18S) microbial communities at each sample site for Fresh vs Advanced Decay. *t* test results are indicated by a single star for significant (p=0.01), two stars for highly significance (p<0.001), 'MS' for marginal significance (p~0.05), and 'NS' for not significant. For soils, the Bonferoni corrected p value for a *t* test with three comparisons was 0.017. For bacterial communities, PD alpha diversity decreased at each sample site between the Fresh and Advanced Decay stages. For microbial eukaryotic communities, changes in PD alpha diversity were variable across sample sites—decreasing for soil, no significant change for the abdominal cavity, and increasing for skin. However, for Shannon diversity (evenness), microbial eukaryotic communities significantly decreased between the Fresh and Advanced Decay stages at every site (data not shown, soil p<0.001, abdominal p=0.002, skin p=0.03).

## Materials and methods

### Experimental set-up and sample collection

We performed a laboratory experiment in which 40 mice were allowed to decompose on soil graves in the University of Colorado Transgenic Facility, which provides a clean, well-controlled environment.

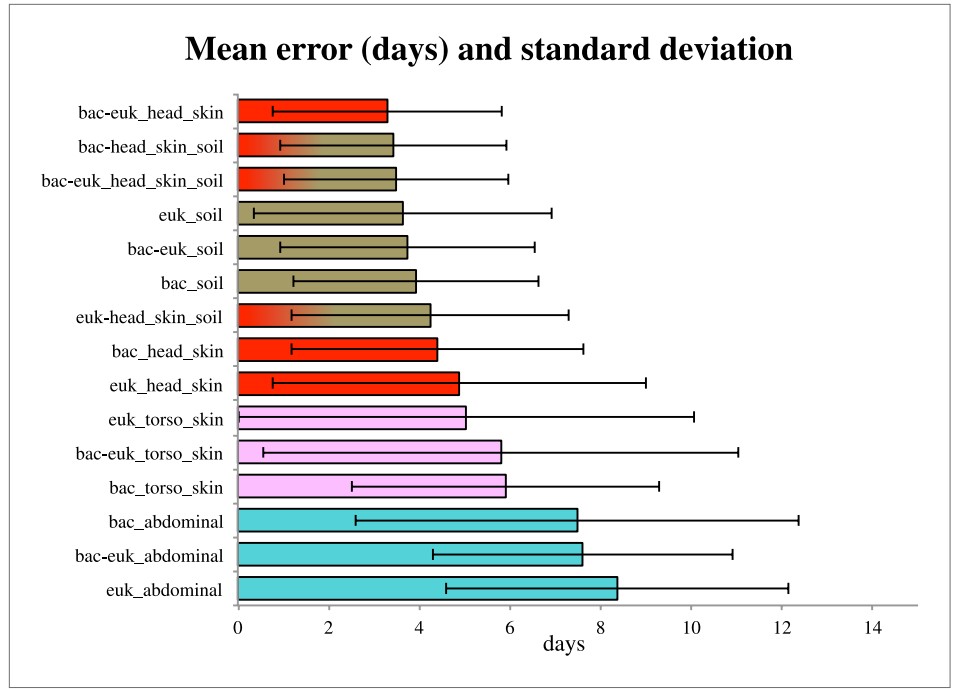

**Figure 5**. Estimates of the mean absolute error (MAE) and standard deviations for PMI regressed directly on the taxon relative abundances (in days) based on bacterial ('bac') and eukaryotic ('euk') microbiome composition for each sampled site— soil (brown), abdominal cavity (teal), skin of the head (red), and skin of the body (pink). Results are show for the timeframe of 0–34 days, which produced smallest errors. Results are displayed with the smallest MAE shown at the top (skin of head with combined 16S and 18S data) and largest error at the bottom (18S abdominal data).

Mice (strain B6C3F1) ranged in age from 36–109 days old and were co-housed by date of birth. To avoid biases by age or relatedness, mice were randomly assigned to one of the eight destructive sampling events. Mice were humanely sacrificed using $CO_2$ gas followed by cervical dislocation and placed on their right side on top of approximately 200–300 × *g* of soil in a clean 500 ml plastic (Tupperware-type) container with a small hole drilled into each side above soil level to prevent anaerobic conditions. Soil (pH ~6) was collected from the organic layer of a dry creek bed in Eldorado Creek near Boulder, Colorado and brought to ~55–60% water holding capacity.

We destructively sampled five mice across eight time points over 48 days with sampling events occurring more frequently (every 3 days) over the first 2 weeks when the rate of decomposition was greatest. At each time point before sampling the five mouse corpses, we recorded a visual body score estimate for the head and torso following *Megyesi et al. (2005)*. The head of mice were scored using the following key: **Fresh**: no discoloration (1 point); **Active Decay** (early decomposition): Discoloration (2 points), Purging of decomposition fluids out of eyes, nose, or mouth (3 points), Bloating of neck and/or face (4 points); **Advanced Decay**: Sagging of flesh (5 points), Sinking of flesh (6 points), Caving in of flesh (7 points), Mummification (8 points). The key for the torso is the same as above except that Bloating of abdominal cavity (3 points) precedes Rupture and/or purging of fluids (4 points). (*Supplementary file 1A*, columns 'KEY_HEAD' and 'KEY_TORSO').

At each time point, microbial communities of corpses were sampled at one internal and two external body sites while still in an aboveground position on the gravesoil. Sterile swabs (BD BBL, CultureSwab, Becton Dickinson, USA) were used to sample the skin of the head near the mouth and the skin of the belly. Swabs were vigorously rubbed across each skin site. To sample the internal abdominal cavity, we utilized two techniques depending on whether the abdominal cavity could hold liquid or not. In early time points (abdominal cavity integrity, day 0–20), we used a 26 G 1 ml syringe with 0.5 ml of saline to flush and extract microbes from the abdominal cavity. For later time points (day 34 and 48), we made an incision into the cavity and used a swab to sample the site. For day 34, we utilized both sampling techniques. We also sampled soil from underneath each mouse corpse during the experiment. Immediately

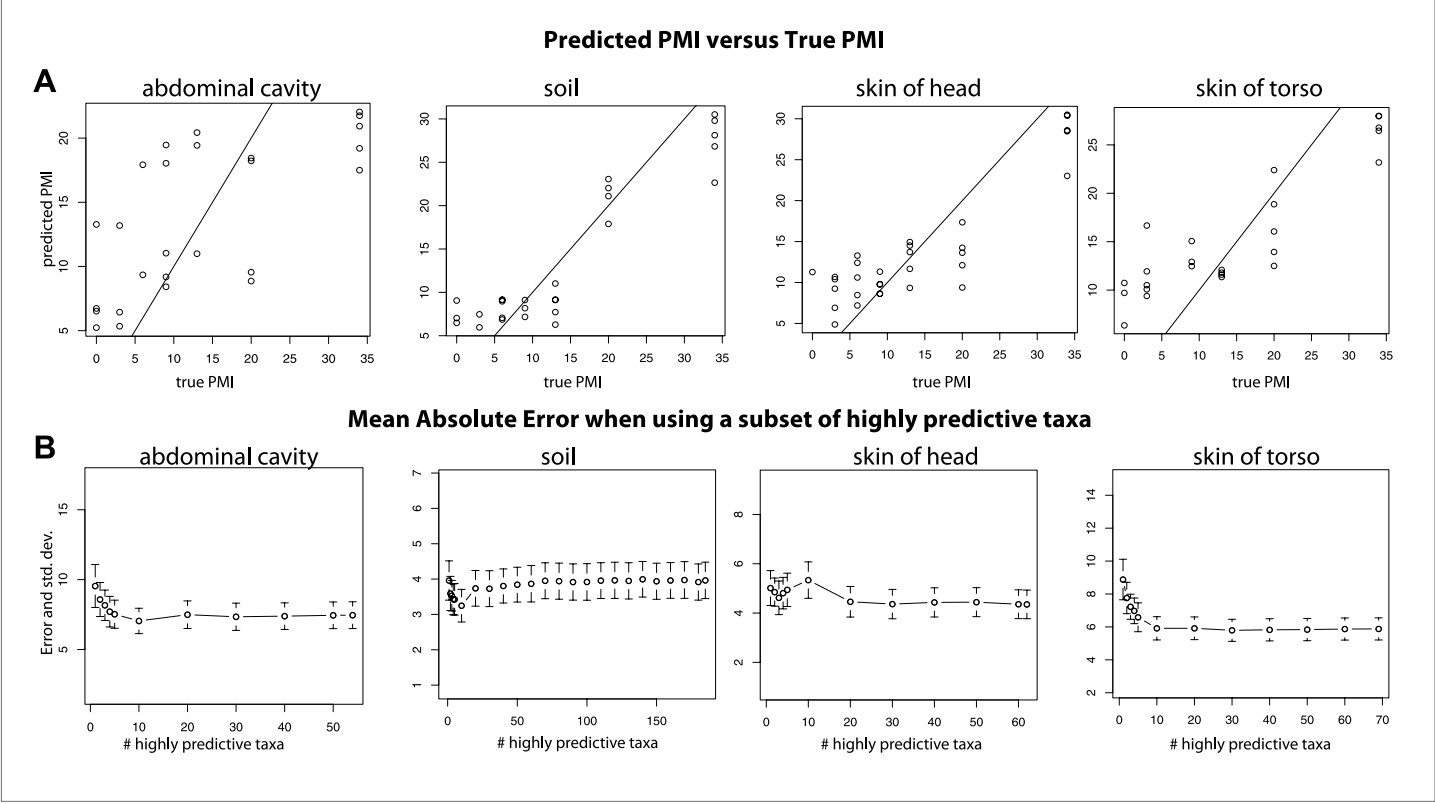

**Figure 6**. Trends of the postmortem interval (PMI) estimates. (**A**) A regression of PMI predictions vs true PMI using 16S data. The line represents a perfect prediction of PMI. The abdominal cavity shows the greatest scatter of points deviating from the line. (**B**) Estimates of the mean absolute error (MAE) for PMI predictions using subsets of highly predictive bacterial taxa. Results suggest that 5–10 highly predictive taxa are required to recover MAEs similar to those generated using the entire microbial community. Similar results were discovered for 18S data, but results are not shown.

after sampling and removing the corpse, surface soil was sampled from under the mouse for a total of four samples associated with each decomposing mouse corpse (or five samples total for mice for which two abdominal samples were taken). Additionally, we collected soil samples at days 0, 6, 20, and 48 from a set of no-corpse control soils that were set up the same as the mice graves (~200–300 × $g$ of soil in a clean 500 ml plastic container). For each soil sample, we suspended 1 × $g$ of each soil sample in 5 ml of deionized water and measured pH using an Orion 3 Star benchtop pH meter (Thermo Scientific, USA). For each sample, triplicate pH measurements were averaged for a final pH estimate for each sample. The mouse corpse and associated samples were frozen immediately after collection and remained at −20°C until further processing.

## DNA extraction, PCR amplification, and next-generation sequencing

DNA extraction, amplicon generation, and amplicon preparation for sequencing followed the protocols recorded in *Caporaso et al. (2011)* and can be found on the Earth Microbiome Project (EMP) webpage (http://www.earthmicrobiome.org/emp-standard-protocols/). Soil samples (~0.30 × $g$), abdominal liquid, and abdominal and skin swabs were extracted using the PowerSoil DNA isolation kit (Mo Bio Laboratories, Carlsbad, CA, USA) at the University of Colorado, Boulder.

For HiSeq Illumina sequencing, total genomic DNA was subjected to PCR amplification (in triplicate) targeting an informative portion of the 16S rRNA variable region 4 (V4) using the bacterial primer set 515F/806R (http://www.earthmicrobiome.org/emp-standard-protocols/16s/). 18S primers were based on *Amaral-Zettler et al. (2009)*, Euk1391f (GTACACACCGCCCGTC) and EukBr (TGATCCTTCTGC AGGTTCACCTAC). A blocking primer specific to mammals was used to minimize amplification of host DNA (GCCCGTCGCTACTACCGATTGGIIIIIITTAGTGAGGCCCT C3 Spacer), with the design based on *Vestheim and Jarman (2008)* as described in the EMP 18S protocol. All primers for amplification and sequencing are available at (http://www.earthmicrobiome.org/emp-standard-protocols/18s/).

For both 16S and 18S, each set of triplicate PCR reactions were subsequently pooled and quantified using Picogreen Quant-iT (Invitrogen, Life Technologies, Grand Island, NY, USA). Negative controls included no template controls for DNA extraction and PCR amplification. Finally, barcoded amplicons were pooled in equal concentrations for each amplicon type (16S and 18S). Each amplicon pool was purified using the Mobio UltraClean PCR Clean-up kit (Mo Bio laboratories, Carlsbad, CA, USA) and sequenced using two lanes of the Illumina HiSeq sequencing platform at the BioFrontiers Institute Next-Generation Genomics Facility at University of Colorado, Boulder.

For sequencing of longer amplicons, we amplified genomic DNA with two primer sets to better determine the taxonomic identity of abundant bacteria and eukaryotes. 16S was targeted with the universal primer set 515f/1391r and 18S was targeted by the primer set 515f/EukBr. PCR amplification and pooling were similar to that described for Illumina sequencing, and detailed primer information and protocols are available on the resources page of qiime.org under long read amplicons (https://s3.amazonaws.com/s3-qiime_tutorial_files/454_FLXplus_protocols_13_4.tgz). Pooled amplicons were sequenced by Pacific Biosciences (Menlo Park, CA, USA). Pooled amplicons were made into SMRTbell libraries followed by the PacBio SMRTbell template preparation protocol and be available at (https://s3.amazonaws.com/files.pacb.com/pdf/Procedure_Checklist_2kb_Template_Preparation_and_Sequencing.pdf). Briefly, after repaired the ends of pooled amplicons, the amplicons then ligate to PacBio SMRTbell adapters to form SMRTbell templates. The templates were then cleaned and purified by Exonuclease treatment and Ampure Bead purification (Agencourt, Beckman Coulter Life Sciences, Indianapolis, IN, USA). The final SMRTbell libraries were bound to PacBioC2 polymerase and sequenced on PacBio RS using C2 sequencing chemistry and 2 × 45 min acquisition time.

## Data analysis

### 16S rRNA sequence processing

Using the default settings in QIIME suite of software tools (*Caporaso et al., 2010*), barcoded Illumina 16S rRNA sequences were quality filtered and demultiplexed using error-correcting Golay codes that reduce the possibility of sample mis-assignment. These reads were 100 bp in length. We classified sequence reads into Operational Taxonomic Units (OTUs) on the basis of sequence similarity. Because our sample set included soil and highly decomposed material, we suspected that a substantial number of sequences would not be represented in the reference Greengenes alignment. Therefore, we utilized the QIIME software open-reference OTU picking protocol. Briefly, sequence reads were initially clustered against the February 2011 release of the Greengenes 97% reference dataset (http://greengenes.secondgenome.com) (*DeSantis et al., 2006*; *McDonald et al., 2012*). Sequences that did not hit the Greengenes dataset were subsequently clustered into de novo OTUs at 97% similarity with UCLUST. Taxonomy was assigned using the RDP classifier (*Cole et al., 2009*) within QIIME retrained on the Greengenes 2011 reference dataset. The representative sequences of all OTUs were then aligned to the Greengenes reference alignment using PyNAST (*Caporaso et al., 2010*) and this alignment was used to construct a phylogenetic tree using FastTree (*Price et al., 2010*) within QIIME. Sequences that did not align to Greengenes with a 70% similarity threshold were assumed to be non-16S and thus artifactual and removed from further analysis. The resulting tree topology with associated branch lengths was used for subsequent diversity analyses. We removed low abundance OTU's making up <0.0005% of reads in the total dataset as recommended for Illumina generated data (*Bokulich et al., 2013*) and removed samples with less than 2500 sequences (for many downstream analyses, samples were rarified at 2500 sequences per sample).

### 18S rRNA sequence processing

Raw 18S rRNA sequence data was subjected to similar processing and demultiplexing protocols within QIIME as described above, except that a curated version of the Silva 108 database was used as the reference database (original: http://www.arb-silva.de/documentation/release-108/; *Pruesse et al. (2007)*, curated version available at qiime.org/home_static/dataFiles.html). Sequences not corresponding to eukaryotic 18S were removed from the dataset prior to analysis by excluding reads that failed to align to the eukaryotic portion of Silva 108 at a low similarity threshold (70% sequence similarity) with PyNAST (*Caporaso et al., 2010*) within QIIME. The dataset was further filtered to exclude all sequences assigned to vertebrate animals, as these likely correspond to the mouse host. Similar to 16S data, we removed low abundance OTU's making up <0.0005% of reads in the total dataset as recommended for Illumina generated data (*Bokulich et al., 2013*) and samples were rarified at 2500 sequences per

sample. A phylogenetic tree was constructed for subsequent diversity analyses by placing representative sequences into a tree of the Silva 108 eukaryotic representative set using the maximum likelihood EPA algorithm within RAxML (**Berger et al., 2011**). Taxonomy was assigned to de novo reads using RDP (**Cole et al., 2009**) retrained on the Silva 108 eukaryotic reference database at the genus level within QIIME.

## Pacific Biosciences data analysis

We took advantages of the long reads provided by the Pacific Biosciences (PacBio) sequencing platform to gain more detailed taxonomic resolution of the abundant bacteria and eukaryotes found in the early and late stage decomposition communities. In both cases high quality circular consensus sequences (CCS) were used. The 16S PacBio data consists of sequences of 800–900 bp in length. The taxonomic identity of the sequences was assessed by assigning sequences to OTUs using the Greengenes February 2011 release of the Greengenes (**DeSantis et al., 2006**; **McDonald et al., 2012**) 97% reference dataset. Taxonomy assignments of the long PacBio reads agreed with the short Illumina reads at the family level and the long reads made is possible to determine the taxonomic identity of the reads to the genus and in most cases species level. Taxonomy of the PacBio sequences was also verified by placing sequences within a phylogenetic tree using RAxML EPA algorithm (**Berger et al., 2011**). The 18S PacBio sequences were roughly 1200 base pairs in length. The 18S data was clustered into OTUs at 97% similarity using the open-reference protocol described above and the curated Silva 108 database. Initial taxonomy assignment was done by BLAST (**Altschul et al., 1990**) with an e-value threshold of e-10. Taxonomy assignment was refined by placing the sequences within the Silva 108 reference tree using maximum likelihood with RAxML EPA. These resulting taxonomy assignments were used to resolve the taxonomy of highly abundant community members. Bacterial and microbial eukaryotic taxa found were sorted by abundance at each site are reported in **Supplementary File 1B**. Genus and species level taxonomy was reported in relevant text of the manuscript. Because PacBio data were generated from only a small subset of samples we did not use these data for comparative analyses, and all statistical analyses were conducted using the Illumina HiSeq data.

## 16S and 18S rRNA HiSeq data analyses

We explored beta diversity patterns by performing principle coordinate analyses (PCoA) with phylogeny-based (UniFrac) unweighted distances. We explored taxon abundance patterns by tracking changes in the abundance of taxa over time using custom python scripts. We determined the most changing taxa by assessing abundance at final time point minus abundance at initial time point (using both percentage, count, and combined measures). We tested whether sample sites during Advanced Decay had similar taxa become abundant by comparing weighted UniFrac distances between sites in each the Fresh stage and the Advanced Decay stage. Alpha diversity was estimated using the phylogenetic diversity metric as well as Shannon diversity. These analyses were performed using the QIIME software. To assess the correlation between microbial community change and time, we performed Mantel tests (with recommended option for single factor tests: unrestricted permutation of raw data) using the software package PRIMER v6 (**Clarke and Gorley, 2006**). Additionally, we tested for significant changes in microbial communities between the Fresh and Advanced Decay stages for each site and data type with a PERMANOVA test in PRIMER v6 (**Clarke and Gorley, 2006**). For statistical analysis, duplicates of samples (e.g., abdominal swab and abdominal liquid/syringe) were removed.

We regressed PMI directly on the taxon relative abundances (derived from the Illumina HiSeq data) using the Random Forests model (**Breiman, 2001**) with version 4.6–7 of the *randomForest* package in *R* (**Liaw and Wiener, 2002**) with default settings. We directly assessed the predictive performance of the regression using leave-one-out (LOO) error estimation, predicting the PMI of each sample using a model trained on the remaining samples. From these LOO errors we calculated the mean error, and standard deviation of the errors with units of days. We performed these regressions within each sample site separately and for bacteria and eukaryotes separately with and without the first and last time points (0 and 48 days) to determine the best body site and timeframe for predicting PMI. Since skin and soil demonstrated the best predictive power ('Results'), we also merged OTU tables from skin and soil sites as well as marker type to potentially further improve our PMI estimates.

Since we sampled the internal abdominal cavity using two techniques because of the inability for the abdominal cavity to hold liquid after ~20–34 days, we controlled for sampling technique. To do this, we sampled day 34 mice using both techniques. Bacterial sequence data generated using both techniques

did not reveal significant difference between communities (PERMANOVA pseudo-F = 0.72, p=0.75) and therefore data generated using both sampling techniques were utilized in downstream analyses.

We found evidence for inhibition in our soil sample DNA extractions. For time points 0–34 days, we extracted DNA from approximately $0.30 \times g$ of soil and found that a 1:20 dilution of DNA resulted in the highest percentage of successful amplifications. For the last time point, we extracted DNA from ~$0.10 \times g$ of soil to reduce inhibition. To assess whether extracting less soil and relieving inhibition pressure would result in a bias such as a change in amplification efficiency for some bacterial groups, we also re-extracted $0.10 \times g$ of soil for a subset of previously collected soil samples (day 0 and day 13). We discovered that the extracting less soil did decrease inhibition, but also resulted in an increase in the relative abundance of actinomycetes across all samples ($0.3 \times g$ vs $0.1 \times g$ soil mean abundance actinomycetes: day 0—mean 0.041 vs 0.26, $t$ test p<0.001; day 13—mean 0.06 vs 0.29 $t$ test p<0.0001). Therefore the day 48 soil 16S sequence data were not directly comparable to 16S sequence data generated for days 0–34 and were excluded from downstream analyses. We did not detect any effect of inhibitions on the results of the 18S data.

## Acknowledgements

DNA sequences have been deposited in the QIIME database as studies 714, 1889, 1996 & 1997 (http://www.microbio.me/emp) and EBI accessions ERP003928, ERP003929,ERP003930, ERP003931. Research was supported by the Office of Justice Programs award 2011-DN-BX-K533, the Earth Microbiome Project (http://www.earthmicrobiome.org/), and Amazon Web Services (AWS) in Education Researchers Grant. We thank the CU Boulder MCDB Transgenic facility. We thank Reece M Gesumaria, J Zelikova, V McKenzie, S Reed, and J Clemente for initial discussions about experiment design. We thank S Song for editorial assistance. We thank J Huntley and the CU next-generation sequencing facility. We thank Jason Chin, Lawrence Lee, James Bullard, and Dimitris Iliopoulos of PacBio for assistance, data processing and comments. We thank N Barger for use of pH meter. Research capacity and infrastructure at Chaminade University of Honolulu is supported by NIH-BRIC P20MD006084.

## Additional information

### Competing interests

YG: YG is an employee of Pacific Biosciences, whose technology is used in this study. JB: JB is an employee of Pacific Biosciences, whose technology is used in this study. The other authors declare that no competing interests exist.

### Funding

| Funder | Grant reference number | Author |
|---|---|---|
| Amazon Web Services (AWS) in Education Researchers Grant | | Rob Knight |
| National Institutes of Justice | 2011-DN-BX-K533 | Jessica L Metcalf, Noah Fierer, David O Carter, Rob Knight |
| Howard Hughes Medical Institute | | Rob Knight |

The funders had no role in study design, data collection and interpretation, or the decision to submit the work for publication.

### Author contributions

JLM, LWP, Conception and design, Acquisition of data, Analysis and interpretation of data, Drafting or revising the article; AG, WVT, Analysis and interpretation of data, Drafting or revising the article; CLL, GCH, MJG, DB-L, KK, Conception and design, Acquisition of data, Drafting or revising the article; DK, Analysis and interpretation of data, Drafting or revising the article, Contributed unpublished essential data or reagents; GA, Conception and design, Acquisition of data; YG, JB, Acquisition of data, Analysis and interpretation of data; NF, RK, Conception and design, Drafting or revising the article; DOC, Conception and design, Analysis and interpretation of data, Drafting or revising the article

### Ethics

Animal experimentation: This work used only dead animals that had been sacrificed as a byproduct of other research, and was deemed not to be animal research by the University of Colorado Institutional Animal Care and Use Committee. Mice were euthanized humanely under approved protocol #08-04-ACK-01, PI Gail Ackermann.

## Additional files

### Supplementary files

• Supplementary file 1. (**A**) Excel file containing detailed metadata for each sample. (**B**) Excel file containing detailed OTU and taxonomy information for PacBio samples. (**C**) Excel file containing means, standard deviation, and two sample *t* test of Weighted Unifrac distances for Fresh versus Advanced Stage decomposition. (**D**) Excel file containing PMI estimates from the regression analyses and associated statistics. (**E**) Excel file containing taxa importance estimates associated with the PMI regression analyses.

### Major dataset

**The following datasets were generated:**

| Author(s) | Year | Dataset title | Dataset ID and/or URL | Database, license, and accessibility information |
|---|---|---|---|---|
| Metcalf JL, Parfrey LW, Gonzalez A, Lauber CL, Knights D, Ackermann G, et al. | 2013 | Metcalf_Mouse_Decomp_2011_16S (Illumina HiSeq), QIIME Study ID 714 | ERP003931; http://www.ebi.ac.uk/ena/data/view/ERP003931 | Publicly available at the EBI European Nucleotide Archive (http://www.ebi.ac.uk/ena/). |
| Metcalf JL, Parfrey LW, Gonzalez A, Lauber CL, Knights D, Ackermann G, et al. | 2013 | Metcalf_Mouse_Decomp_2011_18S (Illumina HiSeq), QIIME Study ID 1889 | ERP003930; http://www.ebi.ac.uk/ena/data/view/ERP003930 | Publicly available at the EBI European Nucleotide Archive (http://www.ebi.ac.uk/ena/). |
| Metcalf JL, Parfrey LW, Gonzalez A, Lauber CL, Knights D, Ackermann G, et al. | 2013 | Metcalf_Mouse_Decomp_2011_18S_PacBio, QIIME Study ID 1997 | ERP003928; http://www.ebi.ac.uk/ena/data/view/ERP003928 | Publicly available at the EBI European Nucleotide Archive (http://www.ebi.ac.uk/ena/). |
| Metcalf JL, Parfrey LW, Gonzalez A, Lauber CL, Knights D, Ackermann G, et al. | 2013 | Metcalf_Mouse_Decomp_2011_16S_PacBio, QIIME Study ID 1996 | ERP003929; http://www.ebi.ac.uk/ena/data/view/ERP003929 | Publicly available at the EBI European Nucleotide Archive (http://www.ebi.ac.uk/ena/). |

**Reporting standards:** Standard used to collect data: the data collected in this study follows the MIMARKS guidelines (http://gensc.org/gc_wiki/index.php/MIMARKS).

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
