## [Decision Letter]

[Editors’ note: although it is not typical of the review process at *eLife*, in this case the editors decided to include the reviews in their entirety for the authors’ consideration as they prepared their revised submission.]

Thank you for sending your work entitled “A microbial clock provides an accurate estimate of the postmortem interval in a mouse model system” for consideration at *eLife*. Your article has been favorably evaluated by a Senior editor, Detlef Weigel, myself (as a member of our Board of Reviewing Editors), and two expert reviewers.

We have discussed our impressions on the manuscript and we have reached a clear consensus. While we all like the data that you have collected and how you have processed it, we feel that the paper is rather “thin” on discussion of the methodology and the implications of the results. In short, we want to see the text extensively revised to offer a more in-depth discussion. Not only of how specific numbers were arrived at, e.g., post-mortem interval estimates, but also of the implications of the work. Also, a more detailed discussion of how the current work represents a major leap beyond [38] would be welcome. As you will see, we all made individual recommendations which, after discussing amongst ourselves, feel need to be addressed. Thus, I have included them below to help guide you in the revisions. Please note that at this point we feel that aside from the revisions, no additional data collection/experiments are deemed necessary. Also, we feel that as it currently stands Figure 5 does not add to the manuscript and should be deleted. Alternatively, if you provide a strong case for leaving it in and revise the manuscript to clarify how it supports your findings and main points, it could ultimately be useful.

Below are the individual comments from the three of us.

*Reviewer 1 (Roberto Kolter)*:

My comments need to be prefaced with a statement that I am not an expert on these sorts of analyses. I am swayed by the general importance of the work and, from my naïve perspective, feel that the work was properly carried out. However, I do have two main concerns that I feel need to be addressed before we make a final decision on this paper. The first is that I feel that the paper is written too much for the experts in the field and not for a broad scientific audience. Expanded explanation of the significance of the methodologies used and, in particular, the data analyses employed would make for a much improved manuscript. Second, I do have a concern that the [38] publication may take some of the wind out of the sails of the current manuscript. The authors do state that their analysis is much more extensive and controlled. It will be important to expand a bit further on that.

*Reviewer 2*:

This is an interesting application of cultivation-independent methods for characterizing successional changes during the microbial decomposition of mice corpses. The data indicate that changes in the skin microbiome after death and gravesoils were more reproducible than succession in the abdomen. The authors do not say much about why this was the case.

Overall, I found that discussion of the study design and findings were “thin” and unsatisfying. I would like to see the manuscript ‘fleshed out’ (no pun intended). Here are examples of questions and issues that the authors might elaborate upon. Why were processes in the abdomen less consistent/reproducible than those that occurred on corpse skin? Would the appearance and abundance of generalist nematodes offer advantages because their numbers might simply reflect total bacteria and be more independent from the specific taxa present? The samples in the time series were from corpses that were ‘incubated’ under controlled conditions – would the outcome change if the environment (climate) was more variable? Were there specific taxa that were particularly good indicators of PMI (and can this approach be tested as a means to improve the usefulness of abdominal samples)? Why are the successional processes on decomposition skin more reproducible than those of the abdomen? Is this likely to be the case with other species? ... and so on. The authors can make this manuscript more interesting than it is in its present form.

*Reviewer 3*:

This manuscript describes how microbial community composition changes through time following death in a mouse model system. The objectives of the study were to characterize changes in bacterial and microbial eukaryotic communities during decomposition, and to determine if these changes could be used to accurately predict time since death (the post-mortem interval (PMI)). Overall I found the manuscript to be a significant contribution to the field. Suggestions for improvement are outlined below.

1) Figures 1 and 3 would benefit from more work. I couldn't locate in the manuscript what all of the time intervals correspond to (3, 6, 9, 12, T5, 24, 48 days?). Given the emphasis of the manuscript is prediction over time, I suggest explicitly marking time on the x axis throughout these figures, and including the point of rupture on all graphs. From an ecology perspective, I would have liked to see all stages of decomposition on the graphs: fresh, bloating, rupture, active & advanced decay. What taxa is denoted by orange in Figure 1? It's not obvious that the skin and soil are similar in late stage decomposition as stated in the manuscript text.

2) Controls. It would be valuable to have more information about the soil controls. How many soil controls were there? Why is the control soil data averaged in 1A? Were there bacterial or microbial eukaryotes that changed significantly with time in the controls?

3) Sequence data. It's unclear how the short and long reads were integrated for the analyses. The manuscript reads as if for both 16s and 18s, OTU classification and phylogenetic inference was carried out separately for the short and long reads. To carry out all downstream analyses of taxon and phylogenetic diversity, what was done? What value was added by including the Pacific Biosciences data? How would the results have changed without those long sequences?

4) PMI estimates. To estimate PMI the authors outline three regression approaches. If I understand correctly, only the results from one approach (regression known PMI against taxon relative abundances) are presented. Why? How, for example, were the stats based on regressing known PMI against the PC1 of a PCoA of unweighted UniFrac distances among samples used? I suggest providing graphs illustrating observed versus predicted PMI values for each site. This would give the reader a visual sense of PMI variation across samples in concert with the predictive performance of the microbial data.

5) Figure 5: this figure doesn't add much value. It's not evident what is being connected to what in the bipartite network. I can see that white and yellow points are more clearly separated in the left versus right figure. The main clustering in the left figure is eukaryote versus bacteria while the patterns in the right figure are more dominated by sample type with the eukaryote/bacteria samples mixed up. These observations don't transparently relate to the one statement in the paper about the figure, “later stage bacterial communities did share more OTU's and taxa than early stage communities”.

---

## [Author Response]

Reviewer 1 (Roberto Kolter):

*My comments need to be prefaced with a statement that I am not an expert on these sorts of analyses. I am swayed by the general importance of the work and, from my naïve perspective, feel that the work was properly carried out. However, I do have two main concerns that I feel need to be addressed before we make a final decision on this paper. The first is that I feel that the paper is written too much for the experts in the field and not for abroad scientific audience. Expanded explanation of the significance of the methodologies used and, in particular, the data analyses employed would make for a much improved manuscript. Second, I do have a concern that the*
[38]
*publication may take some of the wind out of the sails of the current manuscript. The authors do state that their analysis is much more extensive and controlled. It will be important to expand a bit further on that*.

Thank you for these comments: we have rewritten the paper to add material that is interesting to non-specialists, and we have specifically addressed the differences between the [38] paper and the present work, and have modified the Introduction (see paragraph starting “Recently, Pechal et al. (2013) demonstrated…”

Some of the specific differences between the present work and Pechal et al. are that we examine decomposition over a far longer period (weeks rather than days, allowing more stages of decomposition to be examined) and use a much larger sample size (dozens rather than three individuals), allowing a much more in-depth assessment of inter-individual variability and reproducibility of the process. Accordingly, our predictive model is substantially better.

Reviewer 2:

*This is an interesting application of cultivation-independent methods for characterizing successional changes during the microbial decomposition of mice corpses. The data indicate that changes in the skin microbiome after death and gravesoils were more reproducible than succession in the abdomen. The authors do not say much about why this was the case*.

*Overall, I found that discussion of the study design and findings were “thin” and unsatisfying. I would like to see the manuscript ‘fleshed out’ (no pun intended). Here are examples of questions and issues that the authors might elaborate upon*.

We have added a substantial amount of additional text about the study design, sample collection, sequencing methods, analysis methods, results, and discussion. We have also added a new figure to help readers understand the progression of decomposition during our experiment.

*Why were processes in the abdomen less consistent/reproducible than those that occurred on corpse skin? Why are the successional processes on decomposition skin more reproducible than those of the abdomen? Is this likely to be the case with other species*?

We can only speculate at this point about why the changes in microbial communities are more consistent across skin sites and soil than in the abdominal cavity. We think it is worth noting that the environmental change that the abdominal microbial community is exposed to after rupture is much more drastic (anaerobic vs aerobic) than the skin and soil microbial communities experience. Perhaps this dramatic change in environment leads to more stochastic responses by the abdominal microbial community. We have added a brief discussion of this to the manuscript.

*Would the appearance and abundance of generalist nematodes offer advantages because their numbers might simply reflect total bacteria and be more independent from the specific taxa present*?

We did not collect the type of data necessary (bacterial biomass and total count of nematodes) to directly address this question. Related to this question, we found that microbial eukaryotes alone did not perform better in our PMI estimates than bacteria and that the nematode was not always the most important contributor to PMI estimates based on 18S data. The overall method that we are championing here is to use the Tree of Life to the fullest extent. However, we did also include a new analysis indicating that using 5-10 highly predictive taxa provide approximately the same leave-one-out prediction accuracy as models built with all taxa (see new Figure 6). Future studies using contrasting soil types may lend more insight into whether particular taxa are more universal and therefore, potentially useful as taxa-specific forensic indicators.

*The samples in the time series were from corpses that were ‘incubated’ under controlled conditions – would the outcome change if the environment (climate) was more variable*?

Yes, this is very likely, but we do not know yet. The goal for this study was to establish if microbial succession was a fundamental property of decomposition. Future studies will address how much these properties are affected by such parameters as temperature, moisture, etc. We have expanded our discussion of the implications of our experiment.

*Were there specific taxa that were particularly good indicators of PMI (and can this approach be tested as a means to improve the usefulness of abdominal samples)*?

Yes, for each PMI regression we have added results from a test of the “importance” values for the taxa. We list the mean percent increase in mean squared error when the taxon is removed from the predictive model, so higher numbers are better. Additionally, we performed an analysis of models built using small numbers of highly predictive taxa and results indicate that 5-10 taxa provide approximately the same leave-one-out prediction accuracy as models built with all taxa.

*The authors can make this manuscript more interesting than it is in its present form. Other than this I have no substantive concerns*.

Reviewer 3:

*This manuscript describes how microbial community composition changes through time following death in a mouse model system. The objectives of the study were to characterize changes in bacterial and microbial eukaryotic communities during decomposition, and to determine if these changes could be used to accurately predict time since death (the postmortem interval (PMI)). Overall I found the manuscript to be a significant contribution to the field. Suggestions for improvement are outlined below*.

*1)*
Figures 1 and 3
*would benefit from more work. I couldn't locate in the manuscript what all of the time intervals correspond to (3, 6, 9, 12, T5, 24, 48 days?). Given the emphasis of the manuscript is prediction over time, I suggest explicitly marking time on the x axis throughout these figures, and including the point of rupture on all graphs. From an ecology perspective, I would have liked to see all stages of decomposition on the graphs: fresh, bloating, rupture, active & advanced decay. What taxa is denoted by orange in*
Figure 1*? It's not obvious that the skin and soil are similar in late stage decomposition as stated in the manuscript text*.

We have made a number of changes in response to these comments and questions:

In the first two paragraphs of the Results and Discussion, we have substantially expanded our text providing an overview of the experiment and we have added text and data (visual body score data collected during the experiment) to estimate stages of decomposition.

To provide a visual overview of the stages of decomposition, we have added a new Figure 1 (which includes the pH data previously shown in Figure 2). In panel A of this figure, we provide a scatterplot of visual body score results, which align well with soil pH results (panel B). Together data displayed in panels A and B allow us to estimate a general timeline for the progression of decomposition during our experiments.

The likely time point of rupture has been made more prominent and added to all results images in Figures 2 and 3 (16S and 18S results figures). We did not add all stages of decomposition to these figures as we felt it would be too busy.

We have modified text and figures to consistently list sampling intervals in days (0-48) and not in sampling time point (T0-T7).

The orange line in original Figure 1 (now Figure 2) represents taxa in the family *Alcaligenaceae*. We have re-ordered the color legend so that bacterial families appear in the order that they appear in the line graphs. We hope this makes it easier to match the line colors to the bacterial family.

We have added new text and statistics supporting the finding that bacterial communities became more similar in the Advanced Decay stage, although they did not converge. We discovered that similar bacterial groups became abundant in later stage communities. We showed that weighted (abundance) UniFrac distances among sites in the Advanced Decay stage were significantly lower than among sites in the Fresh stage.

*2) Controls. It would be valuable to have more information about the soil controls. How many soil controls were there? Why is the control soil data averaged in 1A? Were there bacterial or microbial eukaryotes that changed significantly with time in the controls*?

Soil controls were originally averaged simply to simplify the 16S and 18S data figures. We have modified both figures to include soil controls shown by time point collected. We have also added methods text describing the collection of control soils and statistical analyses of the microbial communities in the control soils.

*3) Sequence data. It's unclear how the short and long reads were integrated for the analyses. The manuscript reads as if for both 16s and 18s, OTU classification and phylogenetic inference was carried out separately for the short and long reads. To carry out all downstream analyses of taxon and phylogenetic diversity, what was done? What value was added by including the Pacific Biosciences data? How would the results have changed without those long sequences*?

In the Introduction and Methods sections, we have added and reorganized text to clarify the length of sequence reads for each data type and how the different data types were used in analyses.

*4) PMI estimates. To estimate PMI the authors outline three regression approaches. If I understand correctly, only the results from one approach (regression known PMI against taxon relative abundances) are presented. Why? How, for example, were the stats based on regressing known PMI against the PC1 of a PCoA of unweighted UniFrac distances among samples used? I suggest providing graphs illustrating observed versus predicted PMI values for each site. This would give the reader a visual sense of PMI variation across samples in concert with the predictive performance of the microbial data*.

We have streamlined and improved text about the method for estimating PMI. We removed the PCoA-based regression analyses because we thought they were confusing and did not add to the paper. We have also added an additional figure (Figure 6) to help visualize the PMI results.

*5)*
Figure 5*: this figure doesn't add much value. It's not evident what is being connected to what in the bipartite network. I can see that white and yellow points are more clearly separated in the left versus right figure. The main clustering in the left figure is eukaryote versus bacteria while the patterns in the right figure are more dominated by sample type with the eukaryote/bacteria samples mixed up. These observations don't transparently relate to the one statement in the paper about the figure, “later stage bacterial communities did share more OTU's and taxa than early stage communities”*.

We have deleted this figure (previous Figure 5) as requested. We have added an analysis based on the abundance of taxa (weighted UniFrac) and modified text to describe these results.